

# Evaluating the utility of the female-specific mitochondrial *f-orf* gene for population genetic, phylogeographic and systematic studies in freshwater mussels (Bivalvia: Unionida)

Brent M. Robicheau[1,5,*], Emily E. Chase[1,*], Walter R. Hoeh[2], John L. Harris[3], Donald T. Stewart[1] and Sophie Breton[4]

[1] Department of Biology, Acadia University, Wolfville, Canada
[2] Department of Biological Sciences, Kent State University, Kent, United States of America
[3] Department of Biological Sciences, Arkansas State University, Jonesboro, United States of America
[4] Department of Biological Sciences, University of Montreal, Montreal, Canada
[5] Current affiliation: Department of Biology, Life Science Centre, Dalhousie University, Halifax, Canada
[*] These authors contributed equally to this work.

Corresponding author
Sophie Breton,
s.breton@umontreal.ca

## ABSTRACT

Freshwater mussels (order: Unionida) represent one of the most critically imperilled groups of animals; consequently, there exists a need to establish a variety of molecular markers for population genetics and systematic studies in this group. Recently, two novel mitochondrial protein-coding genes were described in unionoids with doubly uniparental inheritance of mtDNA. These genes are the *f-orf* in female-transmitted mtDNA and the *m-orf* in male-transmitted mtDNA. In this study, whole F-type mitochondrial genome sequences of two morphologically similar *Lampsilis* spp. were compared to identify the most divergent protein-coding regions, including the *f-orf* gene, and evaluate its utility for population genetic and phylogeographic studies in the subfamily Ambleminae. We also tested whether the *f-orf* gene is phylogenetically informative at the species level. Our preliminary results indicated that the *f-orf* gene could represent a viable molecular marker for population- and species-level studies in freshwater mussels.

## INTRODUCTION

Freshwater mussels (Bivalvia: Unionida) occur globally, except in Antarctica, with more than 800 estimated species (*Bogan & Roe, 2008*). Despite high diversity, many species are critically imperilled (*Regnier, Fontaine & Bouchet, 2009*; *Lopes-Lima et al., 2017*). Approximately 70% of the ~300 North American species are endangered at some level (*Lopes-Lima et al., 2017*). Freshwater mussels are well recognized for their water filtration capabilities, and for the production of obligate parasitic larvae that metamorphose on fish hosts (*Regnier, Fontaine & Bouchet, 2009*; *Lopes-Lima et al., 2017*). They also possess
an unusual system of mitochondrial transmission called doubly uniparental inheritance (DUI), a characteristic shared with various other bivalves (*Gusman et al., 2016*). DUI is the only exception to the strictly maternal inheritance of mitochondrial DNA (mtDNA) in animals and is characterized by having two types of mtDNA in males (the male-transmitted or M-type mtDNA in germline cells and the female-transmitted or F-type mtDNA in soma), and usually one type (the F-type) in females (*Breton et al., 2007*). DNA divergence between M and F mtDNAs within a single male freshwater mussel can reach >40% (*Doucet-Beaupré et al., 2010*). Moreover, each sex-associated mtDNA contains a novel protein-coding gene in addition to the 13 typical genes involved in ATP production (*m-orf* in M-type and *f-orf* in F-type mtDNA; *Breton et al., 2009*; *Breton, Stewart & Hoeh, 2010*; *Milani et al., 2013*). These genes are among the fastest evolving mt genes in freshwater mussels (*Breton et al., 2011a*; *Breton et al., 2011b*; *Mitchell et al., 2016*). They also have hypothesized roles in the maintenance of DUI and sex determination in bivalves (*Breton et al., 2011a*; *Breton et al., 2011b*), with recent *in silico* analyses supporting such hypotheses (*Mitchell et al., 2016*).

Molecular techniques are commonly used to study freshwater mussels (*Mulvey et al., 1997*; *Krebs, 2004*; *Campbell et al., 2008*) since shell morphology alone is often inadequate to define populations, species or subfamilies. Environmental conditions can affect shell developmental patterns and obfuscate taxonomic identification (*Bogan & Roe, 2008*). Recent divergence (and retention of ancestral morphological characteristics) and hybridization phenomena also make shell characters only partially efficient in discriminating certain populations or lineages (*Hoeh et al., 1995*; *Cyr et al., 2007*). For example, significant genetic differences have been discovered in *Utterbackia* populations with little to no apparent differences in shell morphology (*Hoeh et al., 1995*). Several studies have used both F- and M-type mtDNA sequences. For example, fragments of the 16S rRNA and cytochrome c oxidase subunit I (*cox1*) genes obtained with the universal primers 16Sar-5 and 16Sbr-3 (*Palumbi et al., 1991*) and HCO2198 and LCO1490 (*Folmer et al., 1994*), respectively (or with modified versions of the latter two (*Walker et al., 2006*)), have been used to answer systematic, phylogenetic (*Krebs, 2004*; *Krebs et al., 2013*; *Doucet-Beaupré et al., 2012*) and phylogeographical (*Mioduchowska et al., 2016*) questions about freshwater mussels. Since the M-type mt genomes typically evolve faster than their F-type counterpart in freshwater mussels (*Krebs, 2004*; *Gusman et al., 2016*), relatively older (i.e., species- or family-level) divergences may be tracked more accurately with analyses of the more slowly evolving F-type mtDNA, while analyses of relatively recent (e.g., population-level) divergences may be facilitated by analyses of the faster evolving M-type mtDNA. For example, only the faster evolving male form of the 16S rRNA gene provided strong evidence of geographical isolation among *Pyganodon grandis* populations from the southern region of the Lake Erie watershed (Ohio, USA) (*Krebs, 2004*). However, because the male mtDNA is restricted to the testes, this requires identification of males, and this is impossible with juvenile specimens or with larvae (glochidia). Moreover, the precarious situation of several freshwater mussel species sometimes require non-destructive sampling of animals (e.g., using mantle snips and thus with no access to the M-type mtDNA), which are then returned to the river bottom (*Inoue et al., 2013*).

To explore species boundaries, evolutionary relationships and geographic distribution of freshwater mussel species, researchers also tried other protein-coding loci of the F-type mtDNA such as cytochrome c oxidase subunit II (*cox2*; *Doucet-Beaupré et al., 2012*) and NADH dehydrogenase subunit 1 (*nad1*; *Campbell et al., 2005*). For example, *Campbell et al. (2005)* used *nad1* together with *cox1* and 16S rRNA to study the phylogenetic diversity of the subfamily Ambleminae, but their data could not resolve all the tribes (Amblemini, Lampsilini, Pleurobemini, Quadrulini) as monophyletic assemblages. The same three gene fragments were also found to be poor at resolving recent relationships (intrageneric level) by other researchers (e.g., *Sommer, 2007*; *McCartney et al., 2016*).

Recently, *Wares (2014)* used a straightforward approach and compared whole mitochondrial genome sequences of recently-diverged taxa to identify the most divergent protein-coding region and verify its utility for population genetics (see also *Shearer & Coffroth, 2008*) and systematic studies in scleractinian corals. Although its results suggested that this region alone (cytochrome b, *cytb*) was unlikely to improve researchers' ability to separate coral taxa using DNA sequence-based methods, the proposed pipeline, i.e., to find the most divergent region and to analyze its divergence across available GenBank data, could certainly be adopted to find another useful F-type mitochondrial region for population genetics and systematic studies in freshwater mussels.

Following Wares' pipeline, we focused on Ambleminae, an important freshwater mussel subfamily with several species listed as threatened or endangered (*IUCN, 2015*), and searched for another useful region in the F-type mtDNA to explore phylogenetic diversity and phylogeographic or population genetic structure in this taxa. We compared F mt genomes between two putative species (i.e., two species that are difficult to tell apart morphologically), the Arkansas Fatmucket, *Lampsilis powellii* (I. Lea, 1852) and the Fatmucket, *Lampsilis siliquoidea* (Barnes, 1823) (*Harris et al., 2004*; *Harris et al., 2010*; *Krebs et al., 2013*), and identify highly divergent protein-coding regions such as the *f-orf* gene. We then analyzed sequence divergence in this region (and test whether it is phylogenetically informative) across available amblemine data as a first step to see if it could represent a viable molecular marker for population- and species-level studies in freshwater mussels.

## MATERIALS AND METHODS

*Lampsilis* mussels were collected from two major river drainages in the state of Arkansas: Ouachita River drainage - Ouachita River (Polk County = isolate H2610), and Red River drainage - Mountain Fork Little River (Polk County = isolate H2655). Specimens were obtained under permit, including Arkansas Game and Fish Commission Scientific Collection Permits Nos. 022220078 and 062220101, and Federal Fish and Wildlife Permit No. TE079883-2 issued to JL Harris. Samples were identified as *Lampsilis siliquoidea* or *L. powellii* according to *Harris et al. (2004)* and *Harris et al. (2010)*, i.e., based on external shell morphology (color rays absent, pit rays present, nacre color matte yellow to tan = *L. powellii*; color rays present, pit rays absent, nacre color shiny yellow to tan = *L. siliquoidea*). Each individual was sexed through microscopic examination of gonad tissues. Total DNA was extracted from female mantles to obtain the female-transmitted
**Table 1** Primers pairs used in the amplification of the entire F genomes.

| Mitotype Region (Amplicon size) | Primer name | Primer sequence (5′ to 3′) |
|---|---|---|
| **F genome** | | |
| cox2 –rrnL (~11 kb) | *UNIOCOII.2[a] | CAGTGGTATTGGAGGTATGAGTA |
| | Ambl16SFor[c] | CTGGGTTTGCGACCTCGATGTTGGCTTAGGGAAA |
| cox1 –rrnL (~5.5 kb) | *HCO-700y2[b] | TCAGGGTGACCAAAAAAYCA |
| | Ambl16SRev[c] | TTTCCCTAAGCCAACATCGAGGTCGCAAACCCAG |

Notes.

For primer names: Ambl and *, Amblemine-specific primers.
[a] From *Curole & Kocher (2002)*.
[b] From *Walker et al. (2006)*.
[c] See *Breton et al. (2011b)*.

mtDNA using a QIAGEN DNeasy animal kit following the manufacturer's protocol. Complete mtDNAs were PCR amplified, according to the method of *Breton et al. (2011b)* using primers listed in Table 1. Purified products were sequenced using FLX sequencing (McGill University and Genome Quebec Innovation Centre).

Sequences were assembled with MacVector v10.0 (*Rastogi, 1999*), annotated using MITOS (*Bernt et al., 2013*), and compared to published freshwater mussel mtDNAs. Further assessment of tRNA genes used *tRNAScan-SE* v1.21 (*Lowe & Eddy, 1997*). MUSCLE (*Edgar, 2004*) was used within Geneious v10.0.9 (*Kearse et al., 2012*) to align complete mtDNAs. Nucleotide divergence K(JC) across F-to-F mt genomes was determined with DnaSP v5 (sliding-window = 500 bp; step size = 25 bp) (*Librado & Rozas, 2009*).

DNA alignments of individual genes were produced via MUSCLE in MEGA7 (*Kumar, Stecher & Tamura, 2016*). MEGA7 was used to: determine $p$-distances and $dN/dS$ values ($dN$ = nonsynonymous substitutions/nonsynonymous sites; $dS$ = synonymous substitutions/synonymous sites), calculate $Z$-tests, and generate trees. The best substitution model for each gene was chosen via a model selection test in MEGA7, alignments were manually trimmed to start/end positions without gaps, and Maximum-likelihood (ML) trees were generated with 500 bootstrap replicates (complete deletion was used to account for gaps in ML trees). Bayesian inference trees were produced via BEAST v2.4.6 (*Drummond et al., 2012*), using a Hasegawa-Kishino-Yano model for *f-orf* and a Tamura-Nei evolutionary model for *cox1* (based on BEAST modeltest results), a Yule speciation process, and 80 million Markov chain Monte Carlos steps (sampling every 1000 steps). A 10% burn in was used and resulting trees were compiled into the highest probability topology using TreeAnnotator v1.4 (*Rambaut & Drummond, 2002*). Graphs were produced using ggplot2 within R *Team (2015)*. Specimens used in our phylogenetic analyses and in analysis of *f-orf* and *cox1* sequences variability are described in Table 2. Nomenclature followed *Williams et al. (2017)*. Mitochondrial genomes were deposited in GenBank (accession nos. MF326971 and MF326973).

**Table 2  *Cox1* and *f-orf* sequences used in Ambleminae phylogenies and in analysis of *f-orf* and *cox1* sequences variability.**

| Species | cox1 | | f-orf | |
|---|---|---|---|---|
| | **Accession** | **Reference** | **Accession** | **Reference** |
| *Ellipsaria lineolata* | AY654994 | *Campbell et al. (2005)* | HM849378[a] | *Breton et al. (2011b)* |
| | GU085285[a] | *Boyer et al. (2011)* | – | – |
| | HM849071 | *Breton et al. (2011b)* | – | – |
| *Fusconaia flava* | DQ298537[a] | *Burdick & White (2007)* | HM849380 | *Breton et al. (2011b)* |
| | DQ298538 | *Burdick & White (2007)* | HM849381[a] | *Breton et al. (2011b)* |
| | EF033261 | *Chapman et al. (2008)* | HM849382 | *Breton et al. (2011b)* |
| | HM849073 | *Breton et al. (2011b)* | – | – |
| *Lampsilis ornata* | AF385112 | *Roe, Hartfield & Lydeard (2001)* | AY365193[a] | *Serb & Lydeard (2003)* |
| | AY365193[a] | *Serb & Lydeard (2003)* | – | – |
| *Lampsilis powellii* | HM849075[a] | *Breton et al. (2011b)* | MF326971[a] | This study |
| | – | – | HM849384 | *Breton et al. (2011b)* |
| *Lampsilis siliquoidea* | HM849076 | *Breton et al. (2011b)* | MF326973[*] | This study |
| | – | – | HM849385 | *Breton et al. (2011b)* |
| *Lemiox rimosus* | AY655002[a] | *Campbell et al. (2005)* | – | – |
| | EF033256 | *Chapman et al. (2008)* | – | – |
| | HM849093 | *Breton et al. (2011b)* | – | – |
| *Megalonaias nervosa* | AY655007[a] | *Breton et al. (2011b)* | HM849404[a] | *Breton et al. (2011b)* |
| *Potamilus metnecktayi* | HM849099[a] | *Breton et al. (2011b)* | HM849405[a] | *Breton et al. (2011b)* |
| | – | – | HM849406 | *Breton et al. (2011b)* |
| *Quadrula quadrula* | FJ809750[a] | *Breton et al. (2009)* | FJ809750[a] | *Breton et al. (2009)* |
| | KX853888–KX853982 | *Mathias et al. (2018)* | – | – |
| *Reginaia ebenus* | AY654999 | Unpublished | HM849379[a] | *Breton et al. (2011b)* |
| | HM849072 | *Breton et al. (2011b)* | – | – |
| | KF035133[a] | *Inoue et al. (2013)* | – | – |
| *Cyclonaias houstonensis* | KT285649[a] | *Pfeiffer et al. (2016)* | HM849440[a] | *Breton et al. (2011b)* |
| | – | – | HM849441 | *Breton et al. (2011b)* |
| *Cyclonaias tuberculata* | GU085284[a] | *Boyer et al. (2011)* | HM849376[a] | *Breton et al. (2011b)* |
| | HM849069, HM849070 | *Breton et al. (2011b)* | HM849377 | *Breton et al. (2011b)* |
| *Toxolasma lividum* | AF231756[a] | *Bogan & Hoeh (2000)* | HM849451, HM849452, HM849453, HM849454, HM849455 | *Breton et al. (2011b)* |
| | JF326436 | *Campbell & Lydeard (2012)* | HM849456[a] | *Breton et al. (2011b)* |
| *Toxolasma sp. aff. paulum 1* | HM849131[a] | *Breton et al. (2011b)* | HM849458 | *Breton et al. (2011b)* |
| | HM849133 | *Breton et al. (2011b)* | HM849459, HM849460, HM849461, HM849462, HM849463, | *Breton et al. (2011b)* |
| | – | – | HM849464[a] | *Breton et al. (2011b)* |

**Table 2** (*continued*)

| Species | cox1 | | f-orf | |
|---|---|---|---|---|
| | **Accession** | **Reference** | **Accession** | **Reference** |
| *Toxolasma sp. aff. paulum 2* | HM849129[a] | *Breton et al. (2011b)* | HM849465, HM849466, HM849467, HM849468, HM849469, HM849470, HM849471, HM849472, HM849473 | *Breton et al. (2011b)* |
| | HM849130 | *Breton et al. (2011b)* | HM849474[a] | *Breton et al. (2011b)* |
| | HM849132 | *Breton et al. (2011b)* | HM849475 | *Breton et al. (2011b)* |
| *Toxolasma pullus* | – | – | MF326970[a] | This study |
| *Toxolasma texasiense* | AY655023[a] | *Campbell et al. (2005)* | HM849476 | *Breton et al. (2011b)* |
| | – | – | HM849477[a] | *Breton et al. (2011b)* |
| *Truncilla macrodon* | HM849165[a] | *Breton et al. (2011b)* | HM849478[a] | *Breton et al. (2011b)* |
| | KT285658 | *Pfeiffer et al. (2016)* | | |
| *Venustaconcha ellipsiformis* | EF033260[a] | *Chapman et al. (2008)* | FJ809753 | *Breton et al. (2009)* |
| | – | – | HM849529[a] | *Breton et al. (2011b)* |
| | – | – | HM849530 | *Breton et al. (2011b)* |
| *Villosa iris* | HM849199[a] | *Breton et al. (2011b)* | HM849531 | *Breton et al. (2011b)* |
| | HM849200, HM849201 | *Breton et al. (2011b)* | HM849532[a] | *Breton et al. (2011b)* |
| | – | – | HM849533 | *Breton et al. (2011b)* |

**Notes.**

*Cox1* sizes range from 339 to 1,541 bp. *F-orf* sizes range from 240 to 410 bp. Note that for trees sequences were trimmed to a common start and end position in alignments.

[a] Sequences that were specifically used in Ambleminae phylogenies.

## RESULTS

Mitochondrial genome sizes ($F = 16,043$ and $16,990$ bp for *L. powelli* and *L. siliquoidea*, respectively), gene order and compositions are consistent with those of other freshwater mussels (*Doucet-Beaupré et al., 2010*). We detected one large, potentially species-specific indel: a 51 bp indel in the *cox2/nad3* spacer region between the two F genomes.

Individual gene *p*-distances and *dN/dS* ratios are given in Fig. 1. Our results show, consistent with the degree of nucleotide divergence across mt genes in F genomes (Fig. S1), that the *f-orf* gene is among the least conserved genes in F genomes (Fig. 1). Only *atp8* has a higher *dN/dS* ratio (>0.4) than *f-orf*, although a $Z$-test for selection indicated that the probability of rejecting the null hypothesis of $dN = dS$ (neutrality) for this gene was 0.092.

Phylogenetic analyses focused on the *f-orf* gene because several *f-orf* sequences are available in GenBank compare to *atp8*. Three Bayesian phylogenetic trees were built for Ambleminae using: the *f-orf* gene (Fig. 2A), the standard animal mitochondrial *cox1* DNA barcode (*Hebert, Ratnasingham & De Waard, 2003*; Fig. 2B), and both genes concatenated (Fig. 2C). Corresponding ML analyses are in Fig. S2. The *f-orf* sequences led to better bootstrap values than *cox1*. Moreover, members of *Toxolasma* were grouped as a single clade only in the *f-orf* containing trees. Similar results were obtained with ML analyses. BI

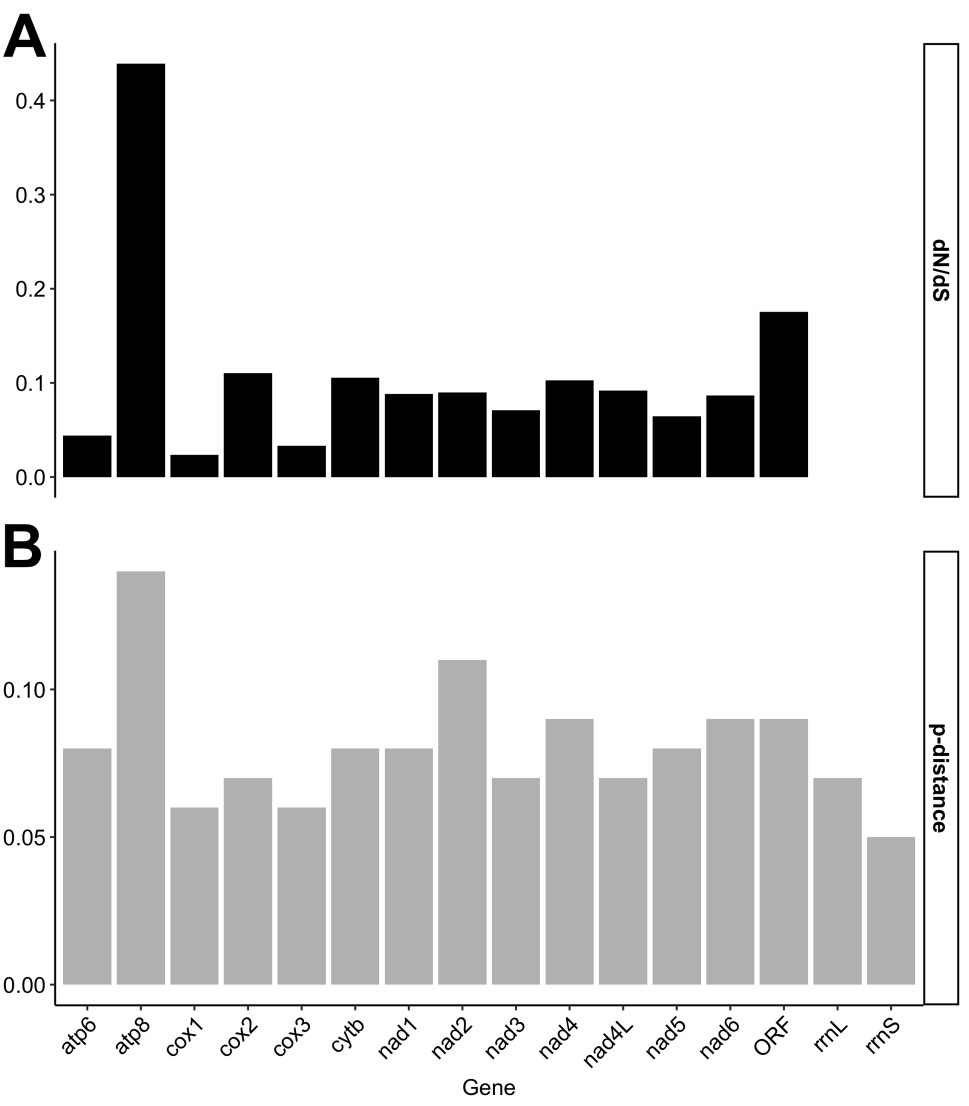

**Figure 1** **Nucleotide distances calculated for individual mitochondrial gene comparisons.** (A) *dN/dS* ratios/scores; (B) *p*-distance scores.

and ML trees with the highest bootstrap values were obtained using both *f-orf* and *cox1* together.

We further examined Ambleminae *f-orf* variability, again using *cox1* for comparison (Fig. 3). For the *f-orf* gene, intraspecific comparisons have lower variability and smaller *p*-distances (range = 0.000–0.011) versus *cox1* (range = 0.000–0.031), and intrageneric comparisons for the *f-orf* gene have overall greater *p*-distances.

## DISCUSSION

Our research objective was to provide guidance towards the identification of another useful region in the F-type mtDNA to explore phylogenetic diversity and phylogeographic

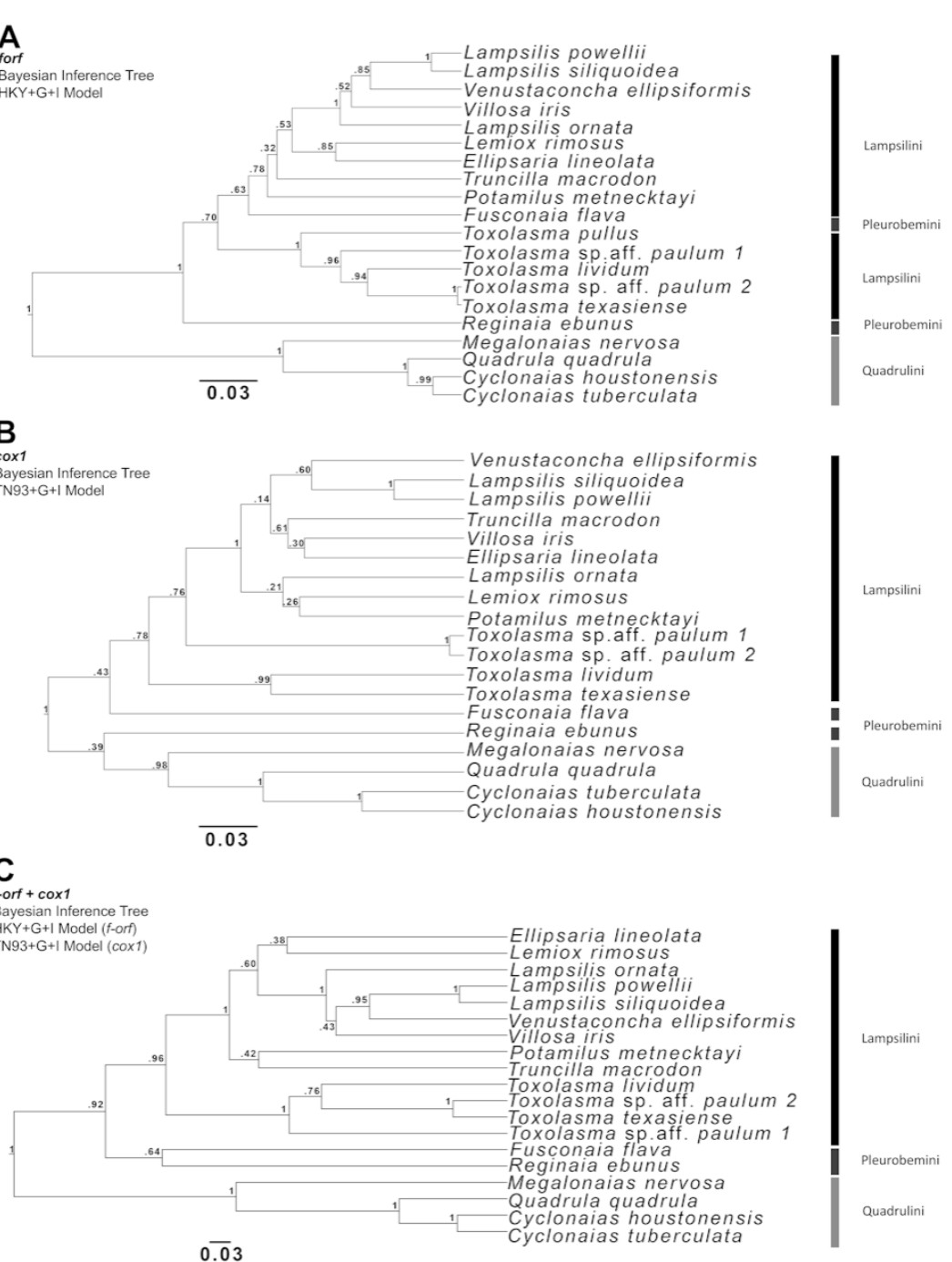

**Figure 2** Bayesian inference (BI) trees produced in BEAUti and BEAST 2.4.6 for (A) *ƒ-orf*, (B) *cox1*, and (C) concatenated *ƒ-orf* and *cox1* sequences also used in maximum likelihood trees of Fig. S2. (A) was produced using a Hasegawa-Kishino-Yano model, (B) a Tamura-Nei model and (C) both these models of nucleotide substitution. Sequences used in BI trees refer to those listed and starred in Table 2.

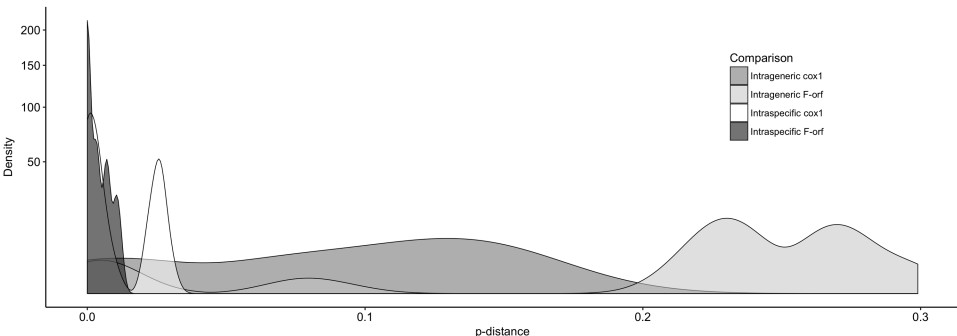

**Figure 3** **The distribution of pair-wise *p*-distance scores for *f-orf* and *cox1* genes within Ambleminae.** The taxa compared are listed in Table 2. Intrageneric comparisons exclude intraspecific comparisons.

or population genetic structure in freshwater mussels. As pointed out by *Wares (2014)*, finding gene regions that provide sufficient information, above and beyond the variation found within a population, might be challenging and we agree with him that it may be more optimal to first explore available genomic data (mitochondrial or whole mtDNA) rather than only use available primer regions or the same gene region that has proven useful in other animal species. In this study, we show that *f-orf*, and also *atp8*, have high divergence between morphologically similar members of *Lampsilis*. Although *atp8* had the highest *dN/dS* ratios, the pattern was consistent with neutrality. This gene has historically been either missing and/or found to be the least conserved in bivalve mtDNAs (*Breton, Stewart & Hoeh, 2010*). Therefore, it is not surprising that it evolves in a different manner than all other mt genes.

Based on divergence data, *f-orf* and *atp8* could thus represent valuable molecular markers in the context of population genetic studies in Ambleminae. Such genes, and in particular *f-orf*, could also potentially help test the hypothetical involvement of mitochondria and their genomes in establishing reproductive barriers and speciation events (*Gershoni, Templeton & Mishmar, 2009*). The general research trend has shown the involvement of sex-linked genes in reproductive isolation (*Qvarnström & Bailey, 2009*), therefore, demonstrating the participation of F- and M-ORF proteins in sex determination, as predicted by *Breton et al. (2011b)*, would particularly corroborate the potential of mitochondrial genetic speciation mechanisms. There were not enough *atp8* sequences (or whole mitochondrial genome sequences) available in GenBank for Ambleminae to see if they could be more informative for population genetic or species delineation studies, however, enough *f-orf* sequences were available to achieve the brief goal of our study.

At the species level, an optimal marker should have a fairly high level of sequence variability, but at the same time it should be sufficiently conserved to reduce phylogenetic noise. Analyses have suggested the systematic usefulness of the *f-orf* gene, while *atp8* appears too noisy. We proceeded with phylogenetic analyses using *f-orf* compared to *cox1* to evaluate the systematic utility of this gene. The *f-orf* BI phylogeny, like *cox1*, was able to distinguish *L. siliquoidea* from *L. powellii* with high bootstrap support. Overall, our data suggest that both genes are somewhat limited in fully resolving Ambleminae phylogenies on

their own, which is illustrated by each tree having relatively low bootstrap values. However, in combination, *f-orf* +*cox1* produced a phylogeny with higher bootstraps, indicating their value for systematic studies.

We further determined *p*-distances in *f-orf* and *cox1* for more sequences within the same lineages. These preliminary data suggest that at the population level, the *f-orf* gene displays relatively fewer nucleotide differences within species, while within genera, there are a relatively larger number of differences (compared to *cox1*). This point is exemplified by *f-orf*s typically having >20% sequence differences between species of the same genus (with the exception of the anomalous *Toxolasma* species pair that had a *p*-distance of <0.05).

## CONCLUSION

This preliminary study indicates that the f-*orf* gene in freshwater mussels could represent a viable molecular marker for population- and species-level studies. This is based on: (1) the *f-orf* gene experiencing a high degree of relaxed purifying selection; (2) the *f-orf* gene, especially when used in combination with *cox1* (and it remains to be seen whether it is the case with other genes), can be phylogenetically informative, and (3) our detection of generally low within-species variability for the *f-orf*, and relatively high between-species variability for most closely related taxa.

## ACKNOWLEDGEMENTS

The following are thanked for providing specimens or assisting with collections: CL Davidson, DM Hayes, WR Posey II and JH Seagraves. We also thank Manuel Lopes-Lima and two anonymous reviewers for insightful criticisms and suggestions.

### Funding

This study was supported by Natural Sciences and Engineering Research Council Discovery Grants awarded to Sophie Breton (RGPIN/435656-2013) and Donald Stewart (RGPIN/217175-2013). The funders had no role in study design, data collection and analysis, decision to publish, or preparation of the manuscript.

### Grant Disclosures

The following grant information was disclosed by the authors:
Natural Sciences and Engineering Research Council Discovery: RGPIN/435656-2013, RGPIN/217175-2013.

### Competing Interests

The authors declare there are no competing interests.

### Author Contributions

- Brent M. Robicheau and Emily E. Chase conceived and designed the experiments, performed the experiments, analyzed the data, contributed reagents/materials/analysis

tools, prepared figures and/or tables, authored or reviewed drafts of the paper, approved the final draft.

- Walter R. Hoeh and John L. Harris conceived and designed the experiments, contributed reagents/materials/analysis tools, authored or reviewed drafts of the paper, approved the final draft.
- Donald T. Stewart conceived and designed the experiments, analyzed the data, contributed reagents/materials/analysis tools, authored or reviewed drafts of the paper, approved the final draft.
- Sophie Breton conceived and designed the experiments, performed the experiments, analyzed the data, contributed reagents/materials/analysis tools, authored or reviewed drafts of the paper, approved the final draft.

## Field Study Permissions

The following information was supplied relating to field study approvals (i.e., approving body and any reference numbers):

Specimens were obtained under permit, including Arkansas Game and Fish Commission Scientific Collection Permits Nos. 022220078 and 062220101, and Federal Fish and Wildlife Permit No. TE079883-2 issued to JL Harris.

## Data Availability

Mitochondrial genomes were deposited in GenBank (accession nos. MF326971 and MF326973).

## Supplemental Information

Supplemental information for this article can be found online at http://dx.doi.org/10.7717/peerj.5007#supplemental-information.

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
