# Peer review of "Evaluating the utility of the female-specific mitochondrial f-orf gene for population genetic, phylogeographic and systematic studies in freshwater mussels (Bivalvia: Unionida)"

_PeerJ, doi:10.7717/peerj.5007_

## Round 0.1 · original submission · Major Revisions

It is my opinion, as well as considering the magnitude of some suggestions made by reviewers that the manuscript requires a substantial revision. Indeed, the presented data are original but the way hypotheses are formulated is not satisfactory and experimental confrontation would benefit from a larger data set, including for marker validation. I suggest refocusing the manuscript in terms of clear aims, and consequent data analyses and discussion. In particular, I concur with one of the reviewers on the suggestion of title change.

Reviewer 1 ·

Basic reporting

no comment

Experimental design

There are certain problems with the definition of research problem, as explained in detail in the General comments.

Validity of the findings

As a consequence of problems with Experimental Design the validity of findings is also questionable, as explained.

Additional comments

Dear Author,
Your manuscript describing the pair of M and F mitogenomes from two Lampsilis species is intriguing.
While reading, I encountered the following problems (referenced by line lumbers):
12: you mentioned population genetic studies yet you report only four new sequences from one locus. This is way below the required >100 from more than one locus.
14-15: the conclusion is not supported by the presented data; figure 2 is hardly promising, with very poor support throughout the tree.
31-33: do you mean freshwater mussels or DUI bivalves in general?
41: the sentence should be split at “because” (the word itself is not needed)
49: italicize “orf”
49-56: the relationship between declared aims and presented data is problematic. You did compare M and F mitogenomes from two (putative?) species – well, at least from four individual representatives of these two species. However, you have not provided any data for population genetics studies, at least nothing serious enough to consider the conclusions. The last objective (phylogenetic utility of f-orf) is not very convincing. You sequenced ca. 30000 bp from each species yet you tested if only 300 bp is usable for phylogenetic analysis? Why? Moreover, you have not used the obtained mitogenomic data in phylogenetic analysis at all… I admit I am confused as this would give you much better context for discussing the possible scenarios leading to the observed polymorphisms. I suggest you re-thing the objectives and focus on the presented new data.
53: check authority
73: how?
83..: I suggest including the whole contents of supplementary material S1 here (both tables, text and references), in the main body of text. Currently the choice of data for comparative analyses is unclear – it is better to have this information in the main body of the ms, particularly if the whole text is rather short and there are no limits on the length.
86: why don’t you use exact numbers?
88-89: the mentioned features are given in the context of which mitogenomes (used as reference)?
90: “are presented”?
97: “are in Genbank” is not acceptable. Also, the explanation in bracket is weak. Please re-think your arguments and rephrase. The real reason behind using this data set is that you previously generated it (this is obvious when looking at Table S2) for your MBE 2011 publication. Very few individual f-orf sequences have been deposited after that, and they are mostly derived from mitogenomic data. This raises the question: why not use the whole mitogenomic data instead? Is it really much cheaper and yet efficient to sequence only f-orf? If you argue it is then why do you present only old data in this context?
121: jargon…
131-139: this paragraph is rather speculative. All mitochondrial markers are linked so looking for positive selection signatures in any particular gene is risky at best. Consider also “draft feedback”...
149-153: the conclusion from this should then be that short fragments of mtDNA are poor phylogenetic markers… which is not very surprising… Why don’t you use the whole mitogenomes??
154-159: while potentially interesting this part is not acceptable. By looking at table S2 I see very few sequences derived from single species. These are more like pairs of sequences than like proper population level data sets. Moreover, it is not clear if cox1 and f-orf sequences are derived from the same individuals. Therefore, the reported difference between intraspecies and intragenus polymorphisms for these two genes may well be random effects.
Figure 1 is way too small, almost unreadable
Table S2: there is an error in accession number – HM84985 should really be HM849385

Good luck with your publication.

·

Basic reporting

The paper is very well written, the bibliography is adequate and provide a good context to all sections.
The figures and tables are also approptiate for the manuscript.
All aims were met with success and properly discussed.

Experimental design

The present manuscript is well inside the scope and aims of PeerJ.
The main aims are well defined and of importance to the field. There is clearly a need for fast evolving markers for population genetics in invertebrates and particularly in freshwater mussels.
The science is sound and well executed with novel methodological approaches. All experimental setup is clear and of high scientific level.

Validity of the findings

The paper is of high scientific value and the data robust and well analysed. Conclusions are well supported.

Additional comments

Dear Authors
This paper presents an interesting study about the potential usage of the recently discovered F- and M- mitochondrial ORFs of Freshwater mussels for population genetics and systematics, with a case study on the subfamily Ambleminae. The paper is very well written, the bibliography is adequate and provide a good context to all sections. The present manuscript is well inside the scope and aims of PeerJ.
The main aims are well defined and of importance to the field. There is clearly a need for fast evolving markers for population genetics in invertebrates and particularly in freshwater mussels.
The science is sound and well executed with novel methodological approaches.

Reviewer 3 ·

Basic reporting

GENERAL COMMENTS

The title should be changed, because it indicates that both sex-specific mitochondrial orf genes were analyzed and this is not true.

The authors write about the use of the orf sequence in taxonomy (orf is a gene indicating high variability), but is it consistent with the morphology-based taxonomy?

The introduction provides information that taxonomy based on shell morphology can be problematic due to phenotypic plasticity. This thread is missing in the discussion.

The introduction lacks data about introgression via genome hybridizations.

In-text citations should be formatted according to the requirements of the journal.

Standardize and consistently use everywhere the same symbols.

Some adjustments need to be performed.
Introduction
36-38: And what is the point of this? Have there been such studies in other mussels?
44-46: specify which F- and M-type DNA sequences were useful (e.g. which genes)
46: add some information in the sentence: “Several studies have used both F- and M-type mtDNA sequences to answer systematic, phylogenetic (Krebs et al., 2004, 2013; Doucet-Beaupré, et al. 2012) and phylogeographical (Mioduchowska et al., 2016) questions about freshwater mussels.”
Mioduchowska M, Kaczmarczyk A, Zając K, Zając T, Sell J. (2016) Gender-associated
mitochondrial DNA heteroplasmy in somatic tissues of the endangered freshwater mussel Unio crassus (Bivalvia: Unionidae): Implications for sex identification and phylogeographical studies. J. Exp. Zool. A Ecol. Genet. Physiol. 325: 610-625.

49-50: „Our aim is to evaluate the potential of sex-specific mitochondrial orf genes for population genetic and systematic studies in freshwater mussels” – this aim is wrongly formulated, and thus not implemented (lack of data for all freshwater mussels and few analyses for Ambleminae, in addition based only on f-orf sequences).

Discussion
Lines 147-149: this information seems to be suitable for the results section, but not discussion
Line 149: add the number of figures where you present ML and BI trees

Conclusion
164: In the line 133 information is provided about positive selection – then what kind of selection is it?

In the supplementary information you wrote: ”Samples were identified as Lampsilis siliquoidea or L. powellii based on geographic location”. Since the introduction provides information about the difficulty of species identification based on morphology, what was your confidence that the specimens were properly determined at the species level? Are the barcoding and orf sequences consistent with the systematics based on morphological features?

I would recommend final proof-reading by an English native speaker before submission. Some examples where the language could be improved:

Introduction
18: except for Antarctica
20: In fact, approximately 70%
22: filtration capabilities, and the production
31: The key feature
32: gene in addition to the 13 typical genes
33: the fastest evolving
39: Molecular techniques are commonly used to study
39: remove “is not uncommon”
40: add comma before “since”
41: add comma after “subfamilies”
41: conditions can affect
42: taxonomic identification
46-47: Therefore, it is justified to hypothesize that mitochondrial genes
47: associated with DUI regulation may be

Materials & Methods
61: Complete mtDNAs were PCR amplified, according to the method of Breton et al. (2011)
61: “in two halves”? Rephrase these unclear sentence, please.
62: primers listed in Table S1

Results
86-87: delate “gene” before order and revise the sentence to “… order of genes and their composition in Lampsilis spp. …”
90: are given in Fig. 1.
90: Our results show, consistently with the degree
92: delate “our results show”
97: add comma after “gene”
97-98: because many f-orf sequences are available in GenBank (each tissue except for sperm will contain f-orfs, making them easier to obtain).
102: delate “The” before ML

Discussion
124-125: delate “protein-coding”
125: genes encoding proteins
126: the role of proteins
126: delate “roles”
129: replace: “As such” with “Therefore”
130: it evolves in a different manner than
131: replace: “Taken together” with “In summary”
132: delate comma after “Ambleminae”
132: and they are ideal
134: replace: “hypothesized” with “hypothetical”
135-139: revise the sentence: “The general research trend has shown the involvement of sex-linked genes in reproductive isolation (Qvarnström and Bailey, 2009), therefore, demonstrating the participation of F- and M-ORF proteins in sex determination, as predicted by Breton et al. (2011), would particularly corroborate the potential of mitochondrial genetic speciation mechanisms.”
141: variability, but at the same time it should be sufficiently conserved to reduce phylogenetic noise
141: analyses have suggested
144: the systematic usefulness
146: delate comma after “support”

In Table S1, please change “Primer” to “Primer name” and “Sequence (5’ to 3’) ” to “Primer sequence (5′-3′)”.

In dendrograms (Figure 2 and Figure S2), species names should be written in italics. Apply in Figure 2 the gray scale to indicate Lampsilini, Pleurobeminijak and Quadrulini as in Figure S2.

Experimental design

Materials & Methods
62: How big was the sample? True, the details are given in Table S1 but here, however, at least concise information should be provided.

Results
98: How many m-orf and f-orf genes are in GenBank? Perhaps it is also worthwhile to conduct such analyses on limited m-orf data, because the title suggests that such analyses have been carried out.

In the supplementary information you wrote: “Total DNA was extracted from male gonads or female mantles” – why did you use different tissues? Please, explain this point.

Validity of the findings

'no comment'

Additional comments

REVIEW OF MANUSCRIPT #24609

I read with interest the manuscript entitled “Using sex-specific mitochondrial orf genes for population genetic and systematic studies in freshwater mussels (Bivalvia: Unionida)”.
I found that the draft included some new findings. However, there should be made some adjustments in order to make the reading logic and more fluid. Please see my comments and suggestions. After modification I hope the draft will be beyond the standard of PeerJ.

---

## Round 0.2 · accepted · Accept

All the comments/suggestions have been satisfactorily addressed.

# Reviewer 1 ·

Basic reporting

I accept the rebuttal

Experimental design

It is acceptable

Validity of the findings

I still doubt it but if you insist this is just a pilot suggesting the possibility then I will not object.

Additional comments

I am disappointed you decided not to discuss the M genomes at all and barely did any mitogenomic comparisons but focused your ms on f-orf only instead. Still, this is your work, not mine...

Reviewer 3 ·

Basic reporting

no comment

Experimental design

no comment

Validity of the findings

no comment

Additional comments

I am satisfied with the corrections made by the Authors. The weak points which were included in the preceding version were removed. Now I find this draft impressive and suitable with the standard of PeerJ.